# LAFR: Efficient Diffusion-based Blind Face Restoration via Latent Codebook Alignment Adapter

## Abstract

Blind face restoration from low-quality (LQ) images is a challenging task that requires not only high-fidelity image reconstruction, but also preservation of facial identity. Although diffusion models like Stable Diffusion have shown promise in generating high-quality (HQ) images, their VAE modules are typically trained only on HQ data, resulting in semantic misalignment when encoding LQ inputs. This mismatch significantly weakens the effectiveness of LQ conditions during the denoising process. Existing approaches often tackle this issue by retraining the VAE encoder, which is computationally expensive and memory intensive. To address this limitation efficiently, we propose **LAFR** (**L**atent **A**lignment for **F**ace **R**estoration), a novel codebook-based latent space adapter that aligns the latent distribution of LQ images with that of HQ counterparts, enabling semantically consistent diffusion sampling without altering the original VAE. To further enhance identity preservation, we introduce a multilevel restoration loss that combines constraints from identity embeddings and facial structural priors. Furthermore, by leveraging the inherent structural regularity of facial images, we show that lightweight finetuning of diffusion prior on just **0.9%** of FFHQ dataset is sufficient to achieve results comparable to state-of-the-art methods, reduce training time by **70%**. Extensive experiments on both synthetic and real-world face restoration benchmarks demonstrate the effectiveness and efficiency of LAFR, achieving high-quality, identity-preserving face reconstruction from severely degraded inputs.

## 1 Introduction

Face image restoration is a critical task within the broader domain of image restoration Hu et al. (2020); Zhao et al. (2022); Yang et al. (2021); Chen et al. (2021); Varanka et al. (2024b), with applications ranging from photo enhancement to facial forensics Wan et al. (2023); Menon et al. (2020); Yu et al. (2024). Unlike general images, facial images exhibit highly structured patterns and identity-sensitive features Varanka et al. (2024a); Hu et al. (2021); Bai et al. (2025), necessitating tailored solutions to ensure structural fidelity and identity preservation. Recent advances in diffusion models Ho et al. (2020); Song et al. (2020; 2021); Rombach et al. (2022) have positioned them as strong generative priors across various restoration tasks, including super-resolution, inpainting, and deblurring. Among these, latent diffusion models have gained popularity for face image reconstruction, where inputs are encoded into a latent space before restoration. However, the VAE encoder in the widely adopted Stable Diffusion Rombach et al. (2022) is trained solely on high-quality (HQ) images. When applied directly to low-quality (LQ) inputs, it produces misaligned latent codes, leading to suboptimal restoration performance Wang et al. (2025); Yingqi et al. (2024).

Existing approaches Chen et al. (2025); Wang et al. (2025); Suin & Chellappa (2024) attempt to correct this by retraining the VAE and incorporating complex alignment modules, but at the cost of increased computational load and inference latency. Moreover, most diffusion models are pretrained on general-purpose datasets such as ImageNet Deng et al. (2009), which limits their direct applicability to face-specific tasks. Adapting these models typically requires large-scale datasets such as FFHQ Karras et al. (2019) or FFHQ-retouched Ying et al. (2023), each comprising 70K high-quality face images. Training on such datasets demands significant time, memory, and compute resources.

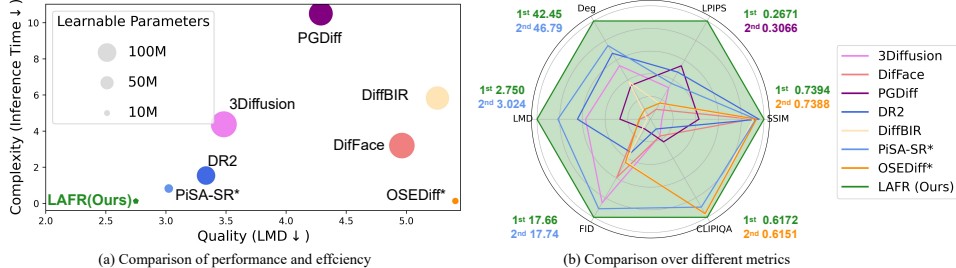

(a) Comparison of performance and effciency      (b) Comparison over different metrics

Figure 1: Comparison of our proposed LAFR with state-of-the-art face restoration methods. (a) Performance and efficiency across different methods. Bubble size reflects the amount of learnable parameters. Our LAFR achieves the best balance, delivering superior quality with minimal computational cost and parameter count. (b) Quantitative comparison across multiple metrics. LAFR outperforms other methods, demonstrating its effectiveness. All metrics are normalized, and for metrics where lower values indicate better performance, we take their reciprocal to ensure consistent visual interpretation across all axes. * means re-trained on FFHQ Karras et al. (2019).

These challenges motivate two central questions: (1) Given the latent space misalignment between LQ and HQ domains, how can the diffusion process be more effectively guided to reconstruct faithful facial details from degraded inputs? (2) Since pretrained diffusion models already capture face-like distributions, can we adapt them for face restoration by updating only a minimal subset of the large-scale training set? In this work, we propose a highly efficient solution that requires just 600 training images (0.9% of FFHQ) and 7.5M trainable parameters to achieve performance comparable to state-of-the-art face restoration methods, saving 70% training time. Our approach effectively bridges the gap between general diffusion priors and the specific demands of facial restoration.

To better leverage facial structure and semantic regularity, we introduce two core innovations. First, we address latent space misalignment with a lightweight codebook-based alignment adapter Van Den Oord et al. (2017); Preechakul et al. (2022); Liang et al. (2025), which transforms LQ latent codes to better align with HQ representations. Second, to ensure robust identity preservation, we introduce a multilevel restoration loss that enforces consistency across appearance, semantic features, identity embeddings, and structural details. Our contributions are summarized as follows:

❏ (1) We propose an efficient strategy that transfers a pretrained diffusion prior to the face restoration task using only a small number of training samples and parameters.

❏ (2) We introduce a codebook-based alignment adapter to resolve the latent space discrepancy between the LQ and HQ face images, facilitating accurate semantic conditioning during sampling.

❏ (3) We design a multilevel restoration loss to maintain facial identity through a combination of appearance, semantics, identity embeddings, and structural alignment.

❏ (4) Extensive experiments on benchmarks for synthetic and real-world face restoration validate the effectiveness and efficiency of our proposed method.

## 2 METHOD

### 2.1 MOTIVATION

The task of restoring a facial image presents unique challenges due to structural regularity and identity Liu et al. (2025); Ying et al. (2024), specific nature of facial features Gao et al. (2025); Liang et al. (2024). A central issue lies in the **latent space misalignment between LQ and HQ images**. In common diffusion-based approaches, such as Stable Diffusion Rombach et al. (2022), the latent encoder, typically a VAE Kingma et al. (2013), is trained exclusively on high-quality images. When LQ inputs are passed through this encoder, the resulting latent representations diverge from the HQ latent distribution, distorting semantic cues that are crucial for accurate reconstruction. This misalignment disrupts the denoising trajectory of diffusion models, as the process is driven by incorrect priors, ultimately compromising the fidelity and consistency of the restored output.

Results in Tab. 7 **Appendix** indicates that directly using LQ latents into the generation produces inferior performance, highlighting the need for an effective alignment strategy to reconcile LQ and

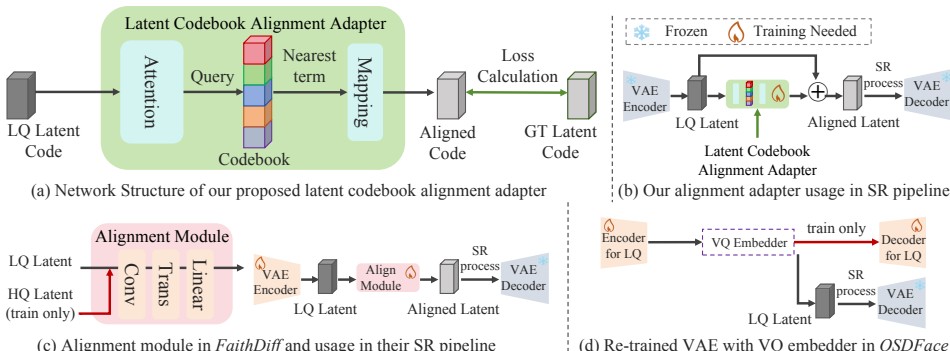

Figure 2: Comparison of alignment strategies of LQ-HQ code in restoration. (a) Our proposed lightweight Latent Codebook Alignment Adapter leverages codebook querying and efficient term mapping, requiring minimal training overhead. (b) Integration of our adapter into our restoration pipeline, where only the adapter is trained. (c) FaithDiff Chen et al. (2025) alignment module uses a parameter-heavy structure involving transformer layers, and needs re-train VAE encoder. (d) OSDFace Wang et al. (2025) retrains a VAE with a VQ embedder and LQ-specific decoder, incurring substantial computational cost. Our method achieves better alignment with significantly reduced training cost and parameter amount.

HQ latent distributions. Without such a mechanism, the diffusion models would not have reached their full potential in face restoration tasks.

At the same time, another important question arises: **How much data is truly necessary to adapt a diffusion model effectively for face restoration**? Most existing approaches are based on large-scale datasets such as FFHQ Karras et al. (2019) and FFHQ-retouched Ying et al. (2023), totaling 140,000 images. This dependency introduces high costs associated with retraining or fine-tuning, which can be prohibitive in practice.

We argue that such a scale may be excessive. Unlike natural images, facial images exhibit strong regularities in geometry and layout across identities. To support this claim, we analyze the feature distributions of face and natural images using ResNet-50 He et al. (2016) embeddings and visualize them via t-SNE Van der Maaten & Hinton (2008). Our analysis reveals that face images (from CelebA Liu et al. (2015)) form a compact and well-defined cluster, while natural images (from ImageNet Deng et al. (2009)) are widely scattered. Quantitatively, face images show a substantially lower intraclass distance (13.28 vs. 24.13) and a higher silhouette score (0.18) Rousseeuw (1987); Shahapure & Nicholas (2020), indicating a high degree of structural coherence. These results suggest that it is feasible to adapt pretrained diffusion models using only a small subset of HQ facial data. Despite variations in pose, lighting, and background, the inherent regularity of facial structure remains intact, offering a compelling opportunity for efficient domain-specific fine-tuning.

## 2.2 OVERVIEW

We propose a diffusion-based framework for blind face restoration, designed for efficient fine-tuning with only a small-scale training dataset. Despite the limited data requirement, our method achieves performance comparable to full fine-tuning on large-scale datasets. Starting from a pretrained diffusion prior, we adapt it to the target domain through an efficient and compact fine-tuning process.

**In the first stage**, we address the domain gap between LQ inputs and HQ training data in latent space by introducing an alignment adapter, illustrated in Fig. 2. This module is trained to project LQ images into the VAE's latent space so that their representations align with those of HQ images. The adapter ensures that the semantic content of the degraded inputs is consistent with the distribution expected by the diffusion model. **In the second stage**, we fine-tune convolutional layers within the denoising UNet of a pretrained Stable Diffusion model. To reduce model complexity, we prune the UNet by removing the text and timestep embedding modules, replacing their output with fixed precomputed tensors. This design significantly reduces the number of trainable parameters and accelerates inference, making our method especially suitable for resource-constrained or data-limited settings. An overview of the complete inference pipeline is provided in Fig. 3. By combining la-

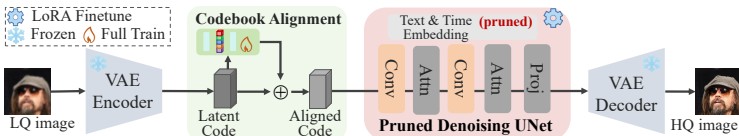

Figure 3: Overall pipeline of our proposed LAFR. We keep VAE encoder and decoder frozen, only training alignment adapter and tuning denoising UNet via LoRA Hu et al. (2022). Tunable parameter groups in UNet are drawn in orange , and frozen parameter groups are drawn in gray .

tent space alignment with efficient UNet LoRA Hu et al. (2022) fine-tuning, our approach produces high-quality, identity-preserving results with minimal training overhead.

## 2.3 LATENT ALIGNMENT ADAPTER FOR LQ IMAGES

In Stable Diffusion, the denoising UNet operates in the latent space, where input images are first encoded into latent codes by a VAE encoder. However, this encoder is trained exclusively on HQ images and fails to generalize to heavily degraded LQ inputs. As a result, LQ images are mapped to semantically distorted latent representations that are misaligned with those of HQ images. This misalignment significantly hinders the effectiveness of the denoising process, often leading to sub-optimal restoration outcomes. Several recent works have attempted to address this issue Li et al. (2020); Suin & Chellappa (2024). OSDFace Wang et al. (2025) retrains a separate visual representation embedder (VRE) module on LQ data and introduces a learnable codebook to enhance semantic alignment. FaithDiff Chen et al. (2025) jointly trains an additional alignment module alongside the VAE encoder and denoising UNet to bring LQ representations closer to their HQ counterparts. While these methods are effective, they often introduce substantial computational overhead or require large-scale training data.

Our goal is to enable efficient domain adaptation using as few additional parameters as possible. To this end, we propose a lightweight Latent Alignment Adapter, inspired by adapter-style architectures. This module aligns LQ latent codes with HQ representations prior to their input to the denoising UNet. The adapter comprises three key components:

**Feature Extractor**. Given an input LQ latent code $z_{LQ} \in \mathbb{R}^{C \times H \times W}$, a shallow convolutional network extracts features $f \in \mathbb{R}^d$. This step captures the core semantics of the degraded input.

**Codebook Matching**. The extracted feature $f$ is used to query a learned codebook Van Den Oord et al. (2017); Preechakul et al. (2022). We perform a nearest-neighbor search to find the closest semantic anchor: $c^* = \arg\min_{c_i \in \mathcal{C}} ||f - c_i||_2$, where each $c_i \in \mathbb{R}^d$ is learned from HQ training data. This step allows the degraded input to borrow high-level semantic guidance from the HQ domain. Due to default $\arg\min$ in PyTorch is not differentiable, we implement a differentiable $\arg\min$ function to propagate gradient correctly, with details in Sec. A.2.2 of **Appendix**.

**Mapping Network**. The matched codebook vector $c^*$ is then passed through a small mapping network $\mathcal{M}$ that projects it back into the latent space, producing the aligned latent code: $z_{aligned} = \mathcal{M}(c^*)$, which is then used as the input to the diffusion model.

To train the alignment adapter, we minimize the distance between the aligned latent code and the ground-truth HQ latent code $z_{HQ}$, obtained by encoding the clean image using the same VAE encoder. The training objective is an $L_1$ loss: $\mathcal{L}_{align} = ||z_{aligned} - z_{HQ}||_1$, which encourages the aligned code to preserve the semantic structure of the HQ face images, even when derived from degraded inputs.

Our alignment adapter introduces minimal computational overhead and a small number of parameters, aligning well with our goal of efficient diffusion model adaptation under limited data and resource constraints. Experimental results in Tab. 8 of **Appendix** demonstrate that this lightweight design significantly boosts both quantitative and qualitative restoration performance.

## 2.4 EFFICIENT FINE-TUNING FOR FACE RESTORATION

After the first training stage mentioned above, which aligns the LQ latent code with those of the HQ images, it is essential to preserve the identity of the input face throughout the restoration process.

Table 1: Quantitative comparison with previous methods, which include non-diffusion, multi-step, and single-step diffusion approaches, with 4× up-scaling. Numbers in brackets indicate denoising steps. Methods named with * mean re-trained on the FFHQ dataset. "RF"++ is short for RestoreFormer++. The best and second-best results of each metric are highlighted in red and blue respectively throughout this paper.

| Method | PSNR↑ | SSIM↑ | LPIPS↓ | DISTS↓ | M-IQ↑ | NIQE↓ | Deg.↓ | LMD↓ | FID-F↓ | FID↓ | C-IQA↑ | M-IQA↑ |
|---|---|---|---|---|---|---|---|---|---|---|---|---|
| GFPGAN Wang et al. (2021) | 25.04 | 0.6744 | 0.3653 | 0.2106 | 59.44 | 4.078 | 34.67 | 4.085 | 52.53 | 42.36 | 0.5762 | 0.5715 |
| RF++ Wang et al. (2023d) | 24.40 | 0.6339 | 0.4535 | 0.2301 | 72.36 | 3.952 | 70.50 | 8.802 | 57.37 | 72.78 | 0.6087 | 0.6356 |
| 3Diffusion (1000) Lu et al. (2024) | 22.32 | 0.6327 | 0.3304 | 0.1716 | 68.54 | 4.334 | 51.05 | 3.482 | 46.26 | 19.45 | 0.5700 | 0.6114 |
| PGDiff (1000) Sun et al. (2025) | 23.87 | 0.6783 | 0.3066 | 0.1794 | 71.70 | 4.883 | 56.07 | 4.289 | 79.38 | 47.01 | 0.5736 | 0.6246 |
| DR2 (300) Sun et al. (2025) | 26.06 | 0.7346 | 0.3131 | 0.2143 | 65.27 | 5.360 | 48.34 | 3.334 | 58.86 | 30.01 | 0.5615 | 0.5989 |
| DifFace (250) Wang et al. (2024b) | 26.47 | 0.7283 | 0.3721 | 0.1676 | 68.98 | 4.154 | 65.84 | 4.963 | 47.76 | 23.65 | 0.5702 | 0.6111 |
| DiffBIR (50) Xinqi et al. (2024) | 27.18 | 0.7293 | 0.3503 | 0.1789 | 74.30 | 5.383 | 55.06 | 5.259 | 68.06 | 27.36 | 0.6168 | 0.6648 |
| PiSA-SR* Sun et al. (2025) | 26.85 | 0.7382 | 0.3254 | 0.1636 | 75.07 | 4.774 | 46.79 | 3.024 | 53.96 | 17.74 | 0.6113 | 0.6701 |
| OSEDiff* Wu et al. (2024a) | 25.88 | 0.7388 | 0.3496 | 0.2278 | 69.98 | 5.328 | 67.40 | 5.408 | 61.36 | 27.13 | 0.6151 | 0.6497 |
| **LAFR** (ours) | 26.26 | 0.7394 | 0.2671 | 0.1792 | 69.99 | 4.715 | 42.45 | 2.750 | 49.20 | 17.66 | 0.6172 | 0.6220 |

Table 2: Comparison on real-world face restoration benchmarks. C-IQA, M-IQA, and M-IQ stands for CLIPIQA, MANIQA, MUSIQ. Methods named with * mean re-trained on the FFHQ dataset. RF++ is short for RestoreFormer++ Wang et al. (2023d).

| Method | Wider-Test | | | | | LFW-Test | | | | | WebPhoto-Test | | | | |
|---|---|---|---|---|---|---|---|---|---|---|---|---|---|---|---|
| | C-IQA | M-IQA | M-IQ | NIQE | FID | C-IQA | M-IQA | M-IQ | NIQE | FID | C-IQA | M-IQA | M-IQ | NIQE | FID |
| GFPGAN | 0.6975 | 0.5205 | 74.14 | 3.570 | 48.57 | 0.6002 | 0.6169 | 73.37 | 4.299 | 47.05 | 0.6696 | 0.4934 | 72.69 | 3.933 | 98.92 |
| RF++ | 0.7159 | 0.4767 | 71.33 | 3.723 | 45.39 | 0.7024 | 0.5108 | 72.25 | 3.843 | 50.25 | 0.6950 | 0.4902 | 71.48 | 4.020 | 75.07 |
| 3Diffusion (1000) | 0.5927 | 0.5833 | 64.24 | 4.489 | 36.39 | 0.6148 | 0.5940 | 68.69 | 4.181 | 47.04 | 0.5717 | 0.5775 | 63.27 | 4.608 | 84.45 |
| PGDiff (1000) | 0.5824 | 0.4531 | 68.13 | 3.931 | 35.86 | 0.5975 | 0.4858 | 71.24 | 4.011 | 41.20 | 0.5653 | 0.4460 | 68.59 | 3.993 | 86.95 |
| DR2 (300) | 0.6544 | 0.5417 | 67.59 | 3.250 | 51.35 | 0.5940 | 0.6088 | 65.57 | 3.961 | 50.79 | 0.6380 | 0.5452 | 66.53 | 4.920 | 97.45 |
| DifFace (250) | 0.5924 | 0.4299 | 64.90 | 4.238 | 37.09 | 0.6075 | 0.4577 | 69.61 | 3.901 | 46.12 | 0.5737 | 0.4189 | 65.11 | 4.247 | 79.55 |
| DiffBIR (50) | 0.7337 | 0.4466 | 72.13 | 4.525 | 36.25 | 0.7266 | 0.4510 | 73.66 | 5.281 | 46.44 | 0.6621 | 0.5799 | 67.50 | 5.729 | 92.04 |
| PiSA-SR* | 0.7265 | 0.5653 | 70.26 | 3.211 | 57.02 | 0.5892 | 0.5942 | 56.35 | 5.957 | 91.95 | 0.6345 | 0.5551 | 62.96 | 4.316 | 107.50 |
| OSEDiff* | 0.6193 | 0.4752 | 69.10 | 5.086 | 47.88 | 0.6186 | 0.4879 | 71.70 | 4.800 | 51.04 | 0.6254 | 0.4823 | 69.81 | 5.325 | 109.23 |
| **LAFR** (ours) | 0.6330 | 0.6099 | 69.51 | 4.859 | 44.69 | 0.6375 | 0.6232 | 69.94 | 3.681 | 45.76 | 0.6956 | 0.6113 | 69.21 | 4.153 | 98.48 |

Traditional restoration losses, such as $L_2$ and LPIPS Zhang et al. (2018), primarily enforce image fidelity and perceptual similarity: $\mathcal{L}_{res} = \|I_{res} - I_{GT}\|_2^2 + \lambda\mathcal{L}_{LPIPS}(I_{res}, I_{GT})$, where $I_{res}$ and $I_{GT}$ denote the restored and ground-truth images, respectively, $\mathcal{L}_{LPIPS}$ is the LPIPS loss function, and $\lambda$ is a balancing weight (set to $\lambda = 2$ in our implementation). Although these losses promote visual consistency, they often fail to capture high-level semantic cues, especially those critical to facial identity. To address this limitation, we design a multilevel loss that incorporates image, identity, and structural supervision to guide the restoration process. This loss consists of the following components:

**Identity Preserving Loss**. To capture high-level identity semantics, we employ a pretrained CLIP vision encoder to extract feature embeddings from both the restored and ground-truth images. The identity loss is then defined as the cosine distance between the two embeddings:

$$\mathcal{L}_{id} = 1 - \cos(\mathcal{F}_{clip}(I_{res}), \mathcal{F}_{clip}(I_{GT})), \tag{1}$$

where $\mathcal{F}_{clip}$ denotes the CLIP Radford et al. (2021) image encoder. This loss encourages the restored face to retain the semantic identity features consistent with the HQ image.

**Facial Structure Loss**. To further enforce structural consistency, we extract spatial structure features and minimize their cosine distance:

$$\mathcal{L}_{fs} = 1 - \cos(\mathcal{F}_{fs}(I_{res}), \mathcal{F}_{fs}(I_{GT})), \tag{2}$$

where $\mathcal{F}_{fs}$ is a dedicated facial structure extractor. This component ensures that detailed structural elements, such as contours, pose, and expressions, are faithfully reconstructed.

The final training objective is a weighted sum of the above components:

$$\mathcal{L}_{total} = \lambda_{res}\mathcal{L}_{res} + \lambda_{id}\mathcal{L}_{id} + \lambda_{fs}\mathcal{L}_{fs}, \tag{3}$$

where we set $\lambda_{res} = \lambda_{id} = \lambda_{fs} = 1$. Detailed ablation of each term is provided in Tab. 5.

A critical design choice in our framework is that we do not use identity embeddings or structural features extracted from LQ inputs as conditioning signals to guide the diffusion process. As discussed in Sec. A.3.2 of **Appendix**, such information is often unreliable due to the severe degradation of LQ

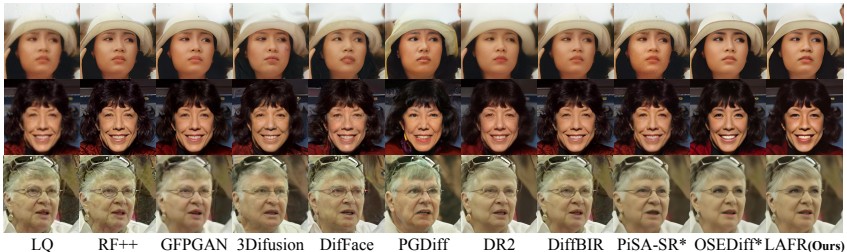

Figure 4: Visualized results on CelebA-Test dataset. RF++ is short for RestoreFormer++ Wang et al. (2023d).

Table 3: Efficiency of our LAFR compared to diffusion-based restoration approaches. Calculation of best and second-best metrics are from Tab. 1, which second-best results are regard as $0.5$. It is noted that pruning has been performed on the UNet in our method. Therefore, despite the introduction of the alignment adapter, the inference time required is still less than that of OSEDiff Wu et al. (2024a).

| Method | Inference Time (s) | Trainable Para. (M) | Training Time (min) | Training Set Size (K) | Best & $2^{nd}$-Best Metric |
|---|---|---|---|---|---|
| PiSA-SR* Sun et al. (2025) | 12.373 | 17.3 | 3516 | 70 | 4.5 |
| PGDiff Yang et al. (2023) | 10.504 | 152.4 | 6083 | 70 | 0.5 |
| DiffBIR Xinqi et al. (2024) | 5.826 | 144.6 | 1822 | 15360 | 2 |
| 3Diffusion Lu et al. (2024) | 4.374 | 180.5 | 2430 | 70 | 1 |
| DifFace Yue & Loy (2024) | 3.197 | 175.4 | 5985 | 70 | 0.5 |
| DR2 Wang et al. (2023c) | 1.217 | 179.3 | 5985 | 70 | 0 |
| OSEDiff* Wu et al. (2024a) | 0.130 | 8.5 | 1440 | 70 | 1 |
| **LAFR** (Ours) | **0.121** | **7.5** | **411** | **0.6** | **5.5** |

images. Conditioning the model on degraded image could lead to superior restoration quality. Instead, by enforcing supervision directly against HQ targets during training, our approach maintains strong visual fidelity while preserving identity information effectively.

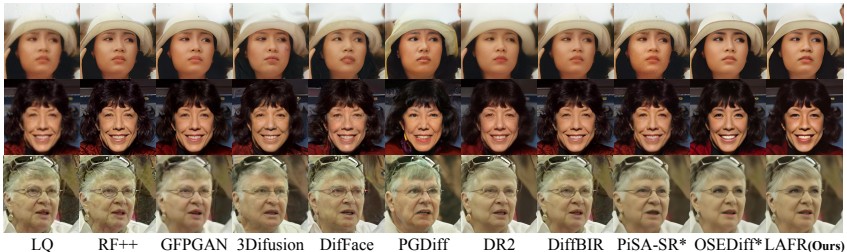

Figure 5: Visualized real-world face restoration. RF++ is RestoreFormer++ Wang et al. (2023d) for short. LQ in each row is selected from WebPhoto-Test, LFW-Test, and Wider-Test, respectively.

## 3 EXPERIMENTS

### 3.1 EXPERIMENTAL SETTINGS

**Datasets**. To evaluate the efficiency of our method under limited supervision, we *randomly* sample 600 images (just 0.9% of the full FFHQ Karras et al. (2019) dataset) as our training set. For evaluation, we use four widely adopted benchmarks: CelebA-Test Liu et al. (2015), WebPhoto-Test Wang et al. (2021), LFW-Test Huang et al. (2008), and Wider-Test Zhou et al. (2022). CelebA-Test consists primarily of synthetically degraded facial images with a $4\times$ downscaling factor, enabling controlled, quantitative evaluation. In contrast, WebPhoto, LFW, and Wider comprise real-world LQ face images captured under diverse and unconstrained conditions, offering a robust testbed for assessing generalization to real-world degradation.

**Evaluation Metrics**. For the synthetic CelebA-Test Liu et al. (2015), we report a comprehensive suite of metrics to assess different aspects of restoration quality, including PSNR, SSIM Wang et al. (2004), LPIPS Zhang et al. (2018), DISTS Ding et al. (2020), MUSIQ Ke et al. (2021), NIQE Zhang et al. (2015), Degree of Degradation (Deg.), Landmark Distance (LMD), FID Heusel et al. (2017) (computed against both FFHQ and ground truth), CLIPIQA Wang et al. (2023a), and

MANIQA Yang et al. (2022). Following prior works Gu et al. (2022); Tsai et al. (2024); Wang et al. (2021), we compute Deg. and LMD using the embedding angle of ArcFace Deng et al. (2019a) to assess identity preservation and fidelity. For real-world benchmarks where ground-truth images are unavailable, we rely on no-reference quality metrics: CLIPIQA, MANIQA, MUSIQ, NIQE, and FID-FFHQ. This combination of full-reference and no-reference metrics enables a comprehensive and balanced evaluation of both objective fidelity and perceptual quality.

**Implementation Details**. For the first-stage training of the alignment adapter, we use the same 600 images mentioned above, with a learning rate of 1e-4, batch size of 16, and 100 epochs. The vocabulary size of the codebook in the latent alignment adapter is set to 1024, with hidden dimension of 256. For the second-stage efficient fine-tuning restoration model, following Wang et al. (2021); Gu et al. (2022); Wang et al. (2025), we build on the pretrained Stable Diffusion v2-1 model Rombach et al. (2022), adopting OSEDiff Wu et al. (2024a) as our baseline diffusion framework. Fine-tuning is preformed with a learning rate of 5e-5, a batch size of 2, and 17,000 total steps. All loss terms are weighted with a coefficient of 1. The identity embedding network $\mathcal{F}_{clip}$ is adapted from IP-Adapter Ye et al. (2023), while the facial structure extractor $\mathcal{F}_{fs}$ is based on D3DFR Deng et al. (2019b). To reduce inference-time parameters, we prune denoising UNet by fixing the prompt to "face, high quality" and removing text and timestep embedding modules, which are precomputed and saved as static tensors, then loaded during inference to minimize model complexity. For efficient fine-tuning, we freeze all layers except those named with "conv" and apply LoRA Hu et al. (2022) with rank of 4 to these layers. All experiments are performed on a single NVIDIA L40 GPU.

## 3.2 Comparison Results

We evaluate the effectiveness of our proposed method on both synthetic and real-world face restoration benchmarks, and compare it against several state-of-the-art approaches.

**Compared Methods**. We compare our method with a range of blind face restoration approaches, including: (1) non-diffusion-based methods: RestoreFormer++Wang et al. (2023d) and GFP-GANWang et al. (2021); (2) multi-step diffusion-based methods: 3Diffusion Lu et al. (2024), PGDiff Yang et al. (2023), DR2 Wang et al. (2023c), DifFace Yue & Loy (2024), and DiffBIR Xinqi et al. (2024); and (3) single-step diffusion-based methods: PiSA-SR Sun et al. (2025) and OSEDiff Wu et al. (2024a). For fair comparison, PiSA-SR and OSEDiff are re-trained on the FFHQ dataset Karras et al. (2019).

**Synthetic Benchmarks**. We use CelebA-Test Liu et al. (2015) as a benchmark, where images are degraded including $4\times$ downsampling, Gaussian blur, JPEG compression, etc. Quantitative comparisons are presented in Tab. 1. Our method achieves comparable or better results than existing methods in terms of metrics, validating the effectiveness of our efficient tuning strategy. Visual comparisons in Fig. 4 further show that our method restores finer facial details and produces more natural-looking outputs than current alternatives, and preserves identity like facial expressions. Beyond accuracy, we also evaluate efficiency. As shown in Tab. 3, our proposed LAFR delivers strong performance while using only 0.9% of the FFHQ training data and 7.5M tunable parameters, reducing training time by 70%, significantly fewer than competing approaches.

**Real-World Benchmarks**. For real-world evaluation, we test on LFW Huang et al. (2008), Wider Zhou et al. (2022), and WebPhoto Wang et al. (2021), which include naturally degraded faces that are unknown and diverse, and without ground-truth references. The results in Tab. 2 show that our method achieves a performance comparable to the state-of-the-art methods, demonstrating a strong generalization to real-world conditions. The qualitative examples in Fig. 5 further illustrate the visual quality of our results. Our method consistently restores vivid, identity-preserving facial details with minimal artifacts, even under severe and unknown degradations. These findings validate the effectiveness of our lightweight design, latent alignment strategy, and identity-consistent supervision in practical face restoration tasks.

## 3.3 Ablation Studies

**Training Set Size**. We evaluate the impact of the training set size by varying the number of FFHQ images used. As shown in Fig. 6, performance improves with more data but exhibits fluctuations under extremely limited supervision. Beyond 600 training images, the improvements become marginal, indicating that a relatively small dataset is sufficient to effectively adapt the diffusion

Table 4: Comparison between (1) our proposed alignment adapter and alignment module in FaithDiff Chen et al. (2025), (2) our proposed multilevel loss and loss in OSDFace Wang et al. (2025), which validates efficiency and performance of our proposal, on Celeb-A. "align." is short for alignment, "Para." for parameter amount (M), and "Infer." for inference time (s). Best results in **bold**.

| Base model | FaithDiff align. | Ours align. | PSNR↑ | FID↓ | NIQE↓ | CLIPIQA↑ | Para.↓ | Infer.↓ |
|---|---|---|---|---|---|---|---|---|
| | ✓ | × | 26.24 | 41.68 | 5.233 | 0.6125 | 2695.9 | 4.956 |
| FaithDiff | × | ✓ | 26.28 | 41.65 | 5.092 | 0.6375 | **2654.6** | **4.064** |
| | ✓ | ✓ | **26.33** | **38.62** | **4.855** | **0.6488** | 2699.4 | 5.276 |
| Ours | ✓ | × | **26.29** | 19.72 | 5.110 | 0.5907 | 1315.8 | 0.136 |
| | × | ✓ | 26.26 | **17.66** | **4.715** | **0.6120** | **1302.0** | **0.121** |

| Base model | OSDFace loss | Ours loss | PSNR↑ | FID↓ | NIQE↓ | CLIPIQA↑ | Para.↓ | Infer.↓ |
|---|---|---|---|---|---|---|---|---|
| Ours | ✓ | × | 24.42 | 26.22 | 6.108 | 0.5244 | **1302.0** | **0.121** |
| | × | ✓ | **26.26** | **17.66** | **4.715** | **0.6120** | 1302.0 | 0.121 |

Figure 6: Performance trend on CelebA-Test as the amount of the training images increases. The model trained on 600 images achieves performance comparable to those trained on larger datasets. prior to the face domain. To further support our observation that a small training set is sufficient to achieve competitive performance in the face restoration task, we conducted additional experiments that evaluated the data efficiency of our approach in Tab. 11 and Tab. 12. We also provide the results of training on overall FFHQ and FFHQR dataset with proposed techniques, in Tab. 9 and Tab. 10 in **Appendix**, to show that this data efficiency is not simple overfitting.

Specifically, we investigate the performance of both OSEDiff Wu et al. (2024a) and our LAFR when trained on subsets of the FFHQ dataset of varying sizes. As shown in Tab. 12 and Tab. 11 in **Appendix**, with significantly reduced training data, both models retain comparable restoration and perceptual quality under data-limited regimes. This supports our claim that (1) high-quality face restoration does not necessarily require large-scale supervision, and (2) our design choices, particularly those related to alignment and latent modeling, contribute to improved sample efficiency. A visualization is shown in Fig. 9 in **Appendix**. Due to page limitations, we provide additional ablation of (1) alignment adapter, (2) denoising UNet pruning, (3) LoRA fine-tuning parameters grouping in Tab. 7 of **Appendix**.

**Loss Function Design**. As shown in Tab. 5, comparing #1 and #2 confirms that adding identity loss improves performance, while #2 vs. #3 shows that cosine distance is more effective than $L_2$ for identity supervision. Similarly, #1 vs. #5 validates the benefit of structure loss, and #4 vs. #5 shows the superiority of its cosine form. The full configuration (#6) achieves the best overall results, demonstrating the complementary effect of combining identity and structure supervision.

Table 5: Ablation study of multi-level loss on CelebA-Test. Each option includes $\mathcal{L}_{res}$. $L_2$ denotes the case includes corresponding loss, but implemented in $L_2$ type.

| # | $\mathcal{L}_{id}$ | $\mathcal{L}_{fs}$ | PSNR↑ | DISTS↓ | NIQE↓ | Deg.↓ | LMD↓ |
|---|---|---|---|---|---|---|---|
| 1 | × | × | 25.99 | 0.1799 | 4.861 | 46.13 | 2.841 |
| 2 | ✓ | × | 26.16 | 0.1872 | 4.921 | 44.87 | 2.821 |
| 3 | $L_2$ | × | 24.75 | 0.1946 | 5.333 | 53.01 | 3.391 |
| 4 | × | $L_2$ | 25.26 | 0.1888 | 4.818 | 49.20 | 3.169 |
| 5 | × | ✓ | **26.28** | 0.1886 | 5.094 | 46.42 | 2.797 |
| 6 | ✓ | ✓ | 26.26 | **0.1795** | **4.716** | **42.45** | **2.750** |

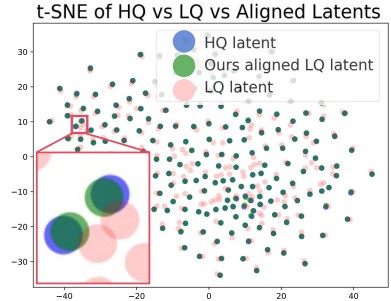

Figure 7: T-SNE result of distribution from LQ, aligned and HQ latent code.

Table 6: Comparison between our proposed alignment adapter and alignment module in VQFR Gu et al. (2022) on CelebA-Test. Best results in red.

| Alignment Model | PSNR↑ | FID↓ | NIQE↓ | CLIPIQA↑ | Para. Amount (M)↓ | Inference Time (s)↓ |
|---|---|---|---|---|---|---|
| VQFR | **26.41** | 27.05 | 5.554 | 0.6105 | 19.0 | 0.128 |
| Ours Alignment Adapter | 26.26 | **17.66** | **4.715** | **0.6120** | **3.5** | **0.121** |

## 3.4 DISCUSSION

**Alignment Adapter**. To evaluate the effectiveness of our alignment adapter, we compare two inference strategies: (1) directly using the latent representation of the LQ image from the VAE encoder, and (2) using the aligned latent representation produced by our alignment adapter. As shown in Fig. 7, our approach significantly reduces the distribution gap between LQ and HQ latent codes (the distance between aligned latent and HQ latent is much closer than that of LQ latent), confirming that the adapter effectively mitigates latent space misalignment.

Recent works such as FaithDiff Chen et al. (2025) also address latent alignment, primarily in general image super-resolution. However, their alignment module introduces larger overhead, incorporating multiple Transformer Vaswani et al. (2017) blocks jointly re-trained with the VAE encoder and fine-tuned alongside the denoising UNet. This integrated, heavyweight design results in increased parameter count, training complexity, and unstable convergence. In contrast, our method is specifically tailored for blind face restoration, where the faces domain features relatively consistent spatial structures and semantic regularity. Leveraging this prior, we design a lightweight alignment adapter based on learned codebook Van Den Oord et al. (2017), effectively bridging the latent gap between LQ and HQ images. This modular design not only minimizes the number of trainable parameters but also enables a two-stage training strategy that decouples alignment from restoration, simplifying optimization. The quantitative comparison between our method and FaithDiff is shown in Tab. 4. Our alignment adapter delivers superior performance with significantly fewer parameters, demonstrating both its practical efficiency and domain-specific effectiveness. Additional comparisons with similar latent alignment methods Kong et al. (2025); Lee et al. (2025); Lin et al. (2023), which all require re-training the VAE encoder, are included in Sec. A.3.1 of **Appendix**.

**Multilevel Loss**. To validate the effectiveness of our multilevel loss, we compare it with the loss function used in OSDFace Wang et al. (2025), which includes DISTS Ding et al. (2020) and an embedding loss derived from ArcFace Deng et al. (2019a). Both losses are applied to our framework under identical training conditions for a fair comparison. As shown in Tab. 4, our multilevel loss consistently achieves superior performance in terms of both identity preservation and perceptual quality. The strength of our approach lies in the ability to supervise the model at multiple semantic levels, including image appearance, identity features, and facial structure. This comprehensive supervision facilitates more balanced learning, enabling the model to preserve facial identity while enhancing visual fidelity. Such a design is particularly well-suited for blind face super-resolution, where both realism and structural integrity are critical.

## 4 CONCLUSION

In this paper, we proposed LAFR, a lightweight and effective framework for blind face restoration via latent space alignment. Motivated by the observation that low-quality (LQ) and high-quality (HQ) face images exhibit a distributional mismatch in the latent space of VAE encodings, we introduce a codebook-guided alignment module that leverages the structural regularity and semantic consistency of facial images to efficiently align LQ features with the HQ latent domain. Building on this, we further introduce a multilevel identity-preserving loss that integrates supervision from image appearance, identity embeddings, and facial structure cues. This design ensures that the restored faces maintain both high perceptual quality and strong semantic identity consistency. With only 600 training images (**0.9%** of FFHQ) and 7.5M trainable parameters, our method effectively adapts a pretrained Stable Diffusion prior to the face restoration task, achieving performance comparable to state-of-the-art methods, also reducing training time by **70%**. Extensive experiments on both synthetic and real-world face restoration benchmarks demonstrate the effectiveness of our approach. Compared to existing latent alignment techniques, LAFR offers a more compact, efficient, and principled solution tailored to the unique characteristics of facial restoration.

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

# A APPENDIX

In the appendix, we demonstrate related work, additional experimental results, implementation details, discussion, and analysis as follows.

## A.1 RELATED WORK

### A.1.1 DIFFUSION-BASED NATURAL IMAGE RESTORATION

Diffusion models Ho et al. (2020); Song et al. (2020; 2021); Rombach et al. (2022) have shown strong potential in image restoration tasks due to their powerful generative priors Dhariwal & Nichol (2021). SR3 Saharia et al. (2022) was among the first to apply diffusion models to the super-resolution task, introducing denoising steps conditioned on low-resolution inputs. Following this, a series of methods explored the use of known degradation operators to guide the diffusion process. DDRM Kawar et al. (2022), DDNM Wang et al. (2023b), and DDS Hyungjin et al. (2024) apply fixed forward operators during sampling to solve inverse problems without retraining. These methods demonstrate strong performance when the degradation process is known. To address scenarios where the degradation type is known but its specific parameters are not, GDP Fei et al. (2023) extends this concept, showing that diffusion models can perform restoration in a task-agnostic manner using generic priors.

For real-world image super-resolution, more recent work has focused on handling complex, unknown degradations. StableSR Wang et al. (2024a), ResShift Yue et al. (2023), and InvSR Yue et al. (2025) adopt multi-step refinement strategies to progressively reconstruct high-quality images from degraded inputs. These methods emphasize intermediate supervision and progressive denoising. In contrast, single-step approaches such as OSEDiff Wu et al. (2024a), SeeSR Wu et al. (2024b), TSD-SR Dong et al. (2025), and FaithDiff Chen et al. (2025) aim to improve sampling efficiency while maintaining the generation quality of the diffusion prior. Further methods, including PiSA-SR Sun et al. (2025) and OFTSR Zhu et al. (2024), explore the controllability between fidelity and realism in the results. These approaches incorporate task-specific designs or losses to ensure semantic accuracy and structural realism.

### A.1.2 DIFFUSION-BASED BLIND FACE RESTORATION

Although general super-resolution methods target natural scenes, several recent works have focused specifically on face restoration using diffusion models Suin et al. (2024); Suin & Chellappa (2024); Chen et al. (2024); Zhao et al. (2023); Qiu et al. (2023); Tu et al. (2022). DR2 Wang et al. (2023c) proposes a diffusion-based face super-resolution framework with domain-specific priors. PGDiff Yang et al. (2023) introduces pose-guided conditioning to enhance the quality of restored

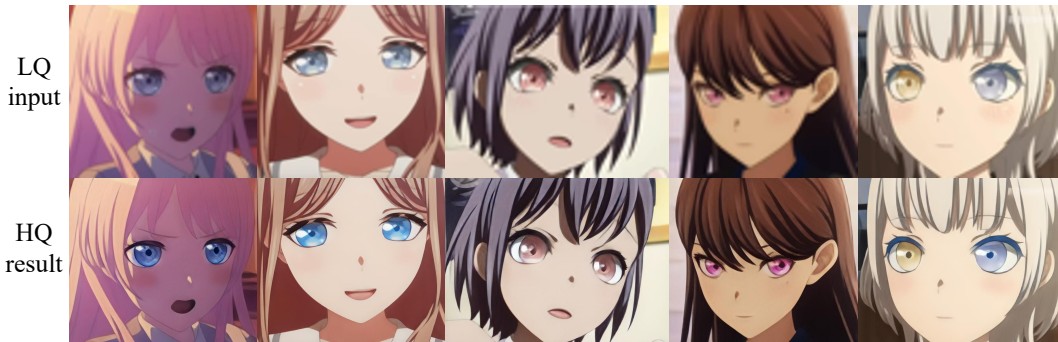

LQ input

HQ result

Figure 8: Results on animation images with real-world degradations. Our method yields high-quality restored images (bottom) given real-world animated low-quality inputs (top).

faces under large pose variations. DifFace Yue & Loy (2024) employs a dual-branch architecture to separate structural and texture generation, improving detail reconstruction. DiffBIR Xinqi et al. (2024) applies a blind image restoration pipeline using diffusion models, specifically designed for real-world face images. These methods demonstrate that the use of facial priors and task-aware guidance can significantly enhance the quality of face-specific restoration Li et al. (2018); Qiu et al. (2023). However, most still depend on full-model fine-tuning or large-scale training datasets. Some related works focus on identity-reserving and 3D-guided face restoration Varanka et al. (2024b); Hu et al. (2020), which provide motivation and inspiration of our proposal.

## A.2 MORE EVALUATION OF LAFR

### A.2.1 MORE EVALUATIONS ON OTHER DOMAINS

Here we provide the restored results on animation images in Fig. 8, which were unseen during our training stages. Degraded animation images are obtained from the Internet, which has degradations including video encoding compressions, severe down-sampling, and other unknown types of degradation. It is shown that (1) our proposed LAFR could generalize to unseen domains, (2) our model is **not over-fitted** on the training set.

### A.2.2 MORE IMPLEMENTATION DETAILS

**Parameters Amount**. Here we provide the detailed calculation of learnable parameters amount. We freeze the VAE encoder and decoder of OSEDiff diffusion model, and for layers in UNet, we set modules with "conv", "conv1", "conv2", "conv_in", "conv_shortcut", "conv_out" trainable, and left frozen. This part has 2.3M learnable parameters in total. For alignment adapter, we set the hidden dimension as 256, and vocabulary size of the codebook as 1024, and the amount of trainable parameters is 5.2M in total. For the overall framework we have 7.5M learnable parameters.

**Differentiable** arg min. We provide the detailed implementation PyTorch-like pseudo code here:

```python
class DifferentiableArgmin(torch.nn.Module):
    def __init__(self, dim, keepdim, beta, straight_through):
        super().__init__()
        self.dim = dim
        self.keepdim = keepdim
        self.beta = beta
        self.straight_through = straight_through

    def forward(self, x):
        return differentiable_argmin(x,self.dim,self.keepdim,self.beta,
        self.straight_through)

class STArgmin(torch.autograd.Function):
    @staticmethod
    def forward(ctx, x, dim, keepdim, beta):
```

```
16          ctx.dim = dim
17          ctx.keepdim = keepdim
18          ctx.beta = beta
19          ctx.save_for_backward(x)
20          return torch.argmin(x, dim, keepdim).to(x.dtype)
21
22      @staticmethod
23      def backward(ctx, grad_output):
24          (x,) = ctx.saved_tensors
25          dim, keepdim, beta = ctx.dim, ctx.keepdim, ctx.beta
26
27          weights = torch.softmax(-beta * x, dim=dim)
28          indices = torch.arange(x.shape[dim])
29          view_shape = [1] * x.dim()
30          view_shape[dim] = x.shape[dim]
31          indices = indices.view(view_shape)
32          soft_idx = torch.sum(weights * indices, dim=dim, keepdim=True)
33          grad_x = -beta * weights * (indices - soft_idx)
34
35          if not keepdim:
36              grad_output = grad_output.unsqueeze(dim)
37
38          return grad_output * grad_x, None, None, None
39
40
41 def differentiable_argmin(x, dim, keepdim, beta, straight_through):
42      if straight_through:
43          return STArgmin.apply(x, dim, keepdim, beta)
44
45      if dim is None:
46          x_flat = x.view(-1)
47          weights = torch.softmax(-beta * x_flat, dim=0)
48          indices = torch.arange(x_flat.shape[0])
49          out = torch.sum(weights * indices)
50          if keepdim:
51              return out.view([1] * x.dim())
52          return out
53
54      weights = torch.softmax(-beta * x, dim=dim)
55      indices = torch.arange(x.shape[dim])
56      view_shape = [1] * x.dim()
57      view_shape[dim] = x.shape[dim]
58      indices = indices.view(view_shape)
59      out = torch.sum(weights * indices, dim=dim, keepdim=keepdim)
60      return out
```

### A.3 MORE DISCUSSIONS

#### A.3.1 COMPARISON WITH OTHER RELATED WORKS

**Comparison with VQFR.** LAFR and VQFR Gu et al. (2022) both incorporate a vector quantization (VQ) codebook as a means of feature modeling. However, the motivations and utilizations of the codebook differ substantially. VQFR leverages the codebook to restore semantically fixed facial structures (e.g., eyes, lips) by learning a representation over Transformer-decoded tokens. In contrast, LAFR applies the codebook to the latent encoding of a VAE, focusing on aligning LQ features directly with their HQ counterparts. Rather than guiding token reconstruction, our codebook acts as an alignment adapter within the VAE latent space. To assess the distinct impact, we replace our alignment adapter with the VQFR codebook module and observe a performance drop (see Tab. 6), confirming the LAFR designed specifically for latent alignment.

**Comparison with CLR-Face.** CLR-Face Suin & Chellappa (2024) similarly addresses the discrepancy between LQ and HQ latent spaces. However, it involves significantly higher computational and training costs. Specifically, CLR-Face first generates a coarse image through an Identity Recovery

Table 7: Ablation studies on CelebA-Test for alignment, UNet pruning, and LoRA parameters. "Pruned" denotes whether the UNet of diffusion model is pruned, "Conv." and "Attn." denotes LoRA finetuning only on convolutional and attention layers.

| # | Align | Pruned | LoRA Type | PSNR↑ | FID↓ | Deg.↓ | NIQE↓ |
|---|-------|--------|-----------|-------|------|-------|-------|
| 1 | × | × | Full | **26.42** | 21.62 | 43.57 | 4.937 |
| 2 | × | × | Attn. | 24.80 | 36.57 | 44.32 | 7.525 |
| 3 | × | × | Conv. | 25.74 | 18.46 | 46.21 | 4.840 |
| 4 | × | ✓ | Conv. | 25.99 | 26.83 | 46.13 | 4.861 |
| **5** | ✓ | ✓ | Conv. | 26.26 | **17.66** | **42.45** | **4.715** |

Network (IRN), which is then refined via a diffusion-based process. The final latent code is subsequently mapped through a VQ codebook and decoded using a VAE decoder. This pipeline requires training multiple components, including the coarse restoration module, VAE encoder, decoder, and the VQ codebook itself. In contrast, LAFR eliminates the need for any coarse preprocessing or full VAE training. Our lightweight alignment adapter efficiently learns to bridge LQ and HQ latent representations without extensive supervision or modular complexity.

**Comparison with SRL-VAE.** SRL-VAE Lee et al. (2025) and LAFR both examine the limitations of conventional VAEs in modeling degraded images, but they differ in goals and solutions. SRL-VAE aims to enhance the robustness of the VAE encoder to corruptions such as blur and watermarking by re-training it on degraded inputs. However, it is not directly applied to downstream restoration tasks. Conversely, LAFR introduces a codebook-based alignment adapter that directly refines the latent space of LQ images toward the HQ domain, enabling effective face image restoration through a subsequent diffusion-based UNet. Our approach emphasizes task-oriented latent alignment rather than encoder robustness alone.

**Comparison with CodeFormer.** While both CodeFormer Zhou et al. (2022) and LAFR utilize a dictionary-learning-based codebook module, their architectural choices and objectives diverge. CodeFormer employs a two-stage training scheme: the first stage learns a codebook prior, and the second stage introduces a Lookup Transformer to retrieve and map latent codes via a learned table-like structure. In contrast, LAFR simplifies this process by learning the codebook and its mapping in a single stage using a lightweight alignment layer. Moreover, CodeFormer enforces direct decoding of the mapped codes into HQ images, whereas LAFR merely aligns the LQ latent distribution to the HQ latent space, deferring image restoration to a subsequent diffusion process. This separation of alignment and restoration enables LAFR to focus on representation-level fidelity rather than immediate pixel-level output.

### A.3.2 EFFECT OF ALIGNMENT ADAPTER, UNET PRUNING, AND LoRA FINE-TUNING

We conduct ablation studies to evaluate the contributions of the alignment adapter, UNet pruning, and LoRA parameter grouping. The results are shown in Tab. 7. Comparing #1 and #2, we observe a minimal performance drop when not fine-tuning attention layers, indicating that attention tuning has limited impact. In contrast, #3 significantly outperforms #2, demonstrating that convolutional layers are more effective for facial structure restoration. Comparison of #3 and #4 shows that UNet pruning introduces negligible degradation, suggesting that the prompt-related modules are redundant. Finally, #5 shows a clear performance gain over #4, confirming the importance of our alignment adapter in improving restoration quality through latent space correction.

### A.3.3 MORE ABLATION STUDIES ON ALIGNMENT ADAPTER

As we mentioned in the motivation part, directly using LQ images or their features as guidance for diffusion sampling could lead to incorrect code, which would further result in an inferior restoration effect. We here provide a possible way of such an approach, which uses IP-Adapter Ye et al. (2023) to extract identity embeddings directly from LQ image as a ControlNet Zhang et al. (2023) guidance for restoration, and its results on the CelebA-Test dataset in Tab. 8. It is shown that our alignment adapter helps achieve better restoration results.

Table 8: Comparison between our alignment adapter and directly using LQ images as conditional guidance via IP-Adapter Ye et al. (2023).

| Method | PSNR↑ | NIQE↓ | FID↓ | LPIPS↓ | Deg.↓ |
|---|---|---|---|---|---|
| Ours w/o alignment w/ IP-Adapter | 26.21 | 5.358 | 25.54 | 0.3606 | 44.74 |
| **Ours** | **26.26** | **4.715** | **17.66** | **0.2671** | **42.45** |

Table 9: Comparison of LAFR trained on 600 FFHQ images, and on overall FFHQ and FFHQR dataset. Results evaluated on CelebA-HQ testset.

| Training set | PSNR | SSIM | LPIPS | DISTS | MUSIQ | NIQE | Deg. | LMD | FID-F | FID | CLIPIQA | MANIQA |
|---|---|---|---|---|---|---|---|---|---|---|---|---|
| 600 images from FFHQ (**Ours**) | 26.26 | 0.7394 | 0.2671 | 0.1792 | 69.99 | 4.715 | 42.45 | 2.750 | 49.20 | 17.66 | 0.6172 | 0.6220 |
| Overall FFHQ and FFHQR | 26.40 | 0.7418 | 0.3454 | 0.1650 | 71.99 | 4.490 | 40.00 | 2.699 | 51.00 | 15.41 | 0.6049 | 0.6344 |

### A.3.4 ABLATION OF TRAINING SET SIZE

A possible question for the observation is that: the results trained on 600 images could be over-fitting, and thus the proposed techniques, including the multi-level loss and alignment adapter, is not making contribution. Here we provide (1) the evaluation results trained on overall FFHQ and FFHQR dataset, which other settings same to ours, in Tab. 9 and Tab.; and (2) the detailed results among different training images amount, where the images are only randomly sampled from FFHQ dataset, in Tab. 11 and Tab. 12.

### A.3.5 LOSS DESIGN

There are some related works using ArcFace as face feature extractor to calculate the loss between the ground truth and the restored result, instead of CLIP used in our loss term $\mathcal{L}_{id}$. The reason for such design is two-folds: (1) CLIP retains rich visual-semantic details (expressions, textures) essential for realistic reconstruction, unlike ArcFace, which prioritizes identity discrimination by suppressing non-identity variations; (2) trained on diverse data, CLIP outperforms ArcFace in handling non-standard scenarios (occlusions, extreme poses), better accommodating reconstruction variability. For detailed implementation of term $\mathcal{L}_{fs}$, we use the 3D facial structure extractor in 3D Morphable Model Lu et al. (2024); Deng et al. (2019b) to calculate the difference between the ground truth and the restored result, in 3D structures. To make sure the gradients are able to be back-propagated, we re-implemented the 3D facial operators used in the extractor via PyTorch.

### A.3.6 WHY NOT USE ALIGNMENT ADAPTER DIRECTLY FOR FACE RESTORATION

Since we have proposed an alignment adapter that receives latent code of LR input and outputs corresponding HR latent code, a question appears: why not just decode the aligned latent code as the restored result, but still need to go through a diffusion model. The reasons for not directly applying adapter for restoration are as follows:

- We only performed alignment at the latent code level, which fails to guarantee the consistency of the images themselves or the validity of restoration.

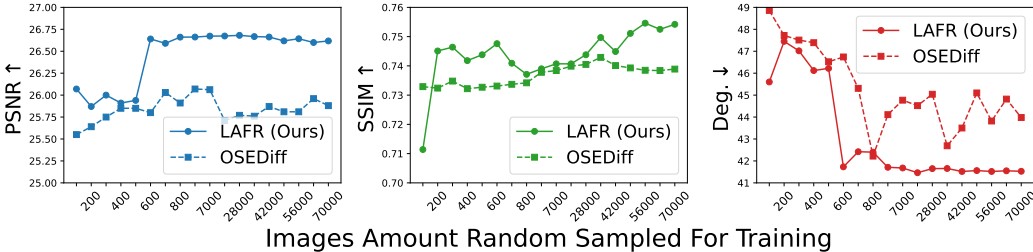

Figure 9: Performance trend of OSEDiff and our LAFR on CelebA-Test, as the amount of training images increases.

Table 10: Comparison of LAFR trained on 600 FFHQ images, and on overall FFHQ and FFHQR dataset. Results evaluated on LFW, Wider, and Webphoto testset. C-IQA, M-IQA and M-IQ are short for CLIPIQA, MANIQA, and MUSIQ separately.

| Training Set | Wider-Test | | | | | LFW-Test | | | | | WebPhoto-Test | | | | |
|---|---|---|---|---|---|---|---|---|---|---|---|---|---|---|---|
| | C-IQA | M-IQA | M-IQ | NIQE | FID | C-IQA | M-IQA | M-IQ | NIQE | FID | C-IQA | M-IQA | M-IQ | NIQE | FID |
| **Ours** | 0.6330 | 0.6099 | 69.51 | 4.859 | 44.69 | 0.6375 | 0.6232 | 69.94 | 3.681 | 45.76 | 0.6956 | 0.6113 | 69.21 | 4.153 | 98.48 |
| Overall | 0.6508 | 0.6336 | 72.13 | 4.135 | 62.15 | 0.6168 | 0.6181 | 71.79 | 4.394 | 57.37 | 0.6179 | 0.6251 | 70.65 | 4.768 | 91.73 |

Table 11: Ablation studies of different training images amount for our LAFR.

| Training Images Amount | PSNR↑ | SSIM↑ | LPIPS↓ | DISTS↓ | MUSIQ↑ | NIQE↓ | Deg.↓ | LMD↓ |
|---|---|---|---|---|---|---|---|---|
| 100 | 26.08 | 0.7384 | 0.2729 | 0.1918 | 71.17 | 5.117 | 46.11 | 3.023 |
| 200 | 26.07 | 0.7114 | 0.2537 | 0.1765 | 71.94 | 4.817 | 45.60 | 2.883 |
| 300 | 25.87 | 0.7451 | 0.2572 | 0.1836 | 71.28 | 4.881 | 47.45 | 2.948 |
| 400 | 25.84 | 0.7425 | 0.2637 | 0.1841 | 71.02 | 4.872 | 46.75 | 2.932 |
| 500 | 26.00 | 0.7464 | 0.2594 | 0.1910 | 71.37 | 5.080 | 47.02 | 2.824 |
| 600 | 25.91 | 0.7418 | 0.2516 | 0.1782 | 71.33 | 4.754 | 46.12 | 2.810 |
| 700 | 25.74 | 0.7388 | 0.2568 | 0.1786 | 72.20 | 4.841 | 46.22 | 3.047 |
| 800 | 26.64 | 0.7476 | 0.2578 | 0.1794 | 68.64 | 4.901 | 41.73 | 2.819 |
| 900 | 26.29 | 0.7309 | 0.2620 | 0.1810 | 69.39 | 4.805 | 42.42 | 2.795 |
| 7000 | 26.43 | 0.7461 | 0.2671 | 0.1831 | 69.37 | 5.044 | 42.87 | 2.773 |
| 21000 | 24.77 | 0.7328 | 0.3862 | 0.1937 | 70.99 | 4.855 | 53.44 | 3.600 |
| 28000 | 25.48 | 0.7497 | 0.3785 | 0.2032 | 66.82 | 5.342 | 51.02 | 3.194 |
| 35000 | 25.56 | 0.7449 | 0.3713 | 0.1962 | 70.64 | 5.183 | 50.56 | 3.176 |
| 42000 | 25.26 | 0.7390 | 0.3735 | 0.1912 | 70.27 | 5.254 | 50.52 | 3.172 |
| 49000 | 25.06 | 0.7371 | 0.3741 | 0.1910 | 71.25 | 4.968 | 52.05 | 3.347 |
| 56000 | 25.27 | 0.7407 | 0.3679 | 0.1826 | 71.50 | 4.953 | 49.65 | 3.144 |
| 63000 | 25.81 | 0.7438 | 0.3623 | 0.1819 | 69.84 | 4.852 | 46.71 | 2.901 |
| 70000 | 25.16 | 0.7380 | 0.3861 | 0.1973 | 70.06 | 5.148 | 52.04 | 3.368 |

- The alignment capability provided by the codebook is also limited; it tends to leverage the distributional similarity of facial images to efficiently learn the feature distribution from LR to HR facial images, rather than engage in direct restoration.

- In fact, when we attempt to directly add restoration-related loss to the VAE and train the model, the VAE transforms into a UNet-like facial restoration model, and numerous studies have shown that the performance of this structure is inferior to that of diffusion-based restoration methods.

### A.3.7 MORE DISCUSSIONS ON MOTIVATION

We have mentioned that one of our motivations is the difference in the distribution between natural and facial images. Due to a compact distribution of facial images, it is possible to use fewer training samples to make a face restoration model learning the corresponding feature than that of natural images. Here we provide a visualized t-SNE result of these two distributions.

Table 12: Ablation studies of different training images amount for OSEDiff.

| Training Images Amount | PSNR↑ | SSIM↑ | LPIPS↓ | DISTS↓ | MUSIQ↑ | NIQE↓ | Deg.↓ | LMD↓ |
|---|---|---|---|---|---|---|---|---|
| 100 | 25.95 | 0.7409 | 0.3719 | 0.1915 | 71.81 | 5.029 | 48.86 | 3.145 |
| 200 | 26.04 | 0.7404 | 0.3607 | 0.1818 | 72.37 | 4.927 | 46.72 | 3.019 |
| 300 | 26.15 | 0.7418 | 0.3592 | 0.1795 | 72.28 | 4.795 | 44.51 | 2.803 |
| 400 | 26.25 | 0.7422 | 0.3524 | 0.1809 | 71.83 | 5.068 | 44.39 | 2.734 |
| 500 | 26.25 | 0.7397 | 0.3545 | 0.1768 | 73.28 | 4.843 | 43.22 | 2.759 |
| 600 | 26.15 | 0.7421 | 0.3560 | 0.1797 | 72.54 | 5.009 | 43.74 | 2.774 |
| 700 | 26.20 | 0.7477 | 0.3460 | 0.1786 | 72.00 | 4.860 | 45.31 | 2.810 |
| 800 | 26.43 | 0.7452 | 0.3472 | 0.1744 | 72.20 | 4.754 | 42.22 | 2.627 |
| 900 | 26.31 | 0.7507 | 0.3478 | 0.1827 | 72.28 | 5.016 | 44.11 | 2.752 |
| 7000 | 26.47 | 0.7514 | 0.3394 | 0.1788 | 72.36 | 5.084 | 44.77 | 2.708 |
| 21000 | 26.11 | 0.7450 | 0.3480 | 0.1779 | 73.39 | 5.130 | 44.52 | 2.771 |
| 28000 | 26.17 | 0.7475 | 0.3387 | 0.1700 | 72.36 | 4.884 | 45.04 | 2.753 |
| 35000 | 26.56 | 0.7489 | 0.3377 | 0.1718 | 71.74 | 4.771 | 42.69 | 2.583 |
| 42000 | 26.57 | 0.7531 | 0.3397 | 0.1759 | 71.58 | 4.906 | 43.50 | 2.640 |
| 49000 | 26.21 | 0.7480 | 0.3511 | 0.1821 | 72.63 | 5.010 | 45.10 | 2.874 |
| 56000 | 26.21 | 0.7465 | 0.3402 | 0.1713 | 72.20 | 4.874 | 43.82 | 2.717 |
| 63000 | 26.36 | 0.7490 | 0.3428 | 0.1778 | 72.50 | 4.908 | 44.82 | 2.782 |
| 70000 | 26.85 | 0.7382 | 0.3254 | 0.1636 | 75.07 | 4.774 | 46.79 | 3.024 |

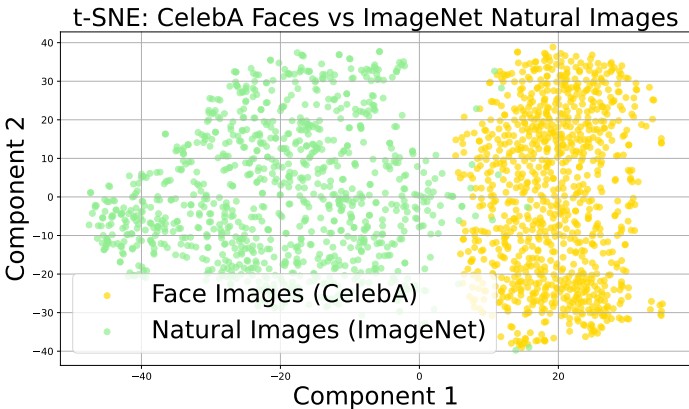

Figure 10: T-SNE results show that compared to natural images, facial images exhibit a more compact distribution.

## A.4 LIMITATIONS

This paper's codebook-based design for alignment is built upon the assumption that facial images share highly similar layouts and distributions. Therefore, semantic alignment between LQ and HQ images can be achieved through dictionary learning in a lightweight structure. However, for complex situations such as extreme poses and diverse natural images, this approach may not produce the same effectiveness. We plan to explore this issue in future work.

## A.5 SOCIETAL IMPACTS

Our efficient face image restoration method can help in medical diagnosis and forensic investigations by restoring faces from partial or degraded data, thus supporting identity verification and facial analysis in critical scenarios. Technology can also benefit individuals with disabilities by enabling more accurate facial expression recognition and avatar reconstruction, enhancing communication in virtual environments. However, the method may be misused for unauthorized reconstruction of individuals' faces, raising concerns about privacy invasion, surveillance, and identity fraud if not properly regulated.

## A.6 LLM USAGE DECLARATION

In the preparation of this document, we utilized Large Language Model (LLM) to enhance the quality of the writing. Its application is focused on text polishing, grammar correction, and improving clarity. All content generated with the assistance of the LLM was rigorously reviewed, revised, and ultimately approved by the authors to ensure its accuracy and originality.

