# OpenReview forum: "LAFR: Efficient Diffusion-based Blind Face Restoration via Latent Codebook Alignment Adapter"
_ICLR.cc/2026/Conference — ICLR 2026 Conference Withdrawn Submission_

### Official Review · Reviewer_UZbd · 2025-10-27

**Soundness:** 2
**Presentation:** 3
**Contribution:** 2
**Rating:** 2
**Confidence:** 5

**Summary:**

The paper presents LAFR, a lightweight latent alignment framework for BFR that bridges the distributional gap between low- and high-quality facial representations without retraining the VAE. By integrating a codebook-guided alignment adapter, multilevel identity–structure loss, and LoRA-efficient diffusion fine-tuning, LAFR delivers state-of-the-art reconstruction fidelity and identity preservation in a single forward pass. Comprehensive evaluations on CelebA-Test, LFW-Test and WebPhoto-Test show consistent improvements in SSIM, PSNR, LMD and CLIPIQA while requiring only 0.9 % of the FFHQ data and 7.5 M trainable parameters.

**Strengths:**

1.	Employs a codebook that has been empirically validated to work well.
2.	Raises the intriguing hypothesis that the VAE fails to align LQ latent codes.

**Weaknesses:**

1.	It is inappropriate to claim that IP-Adapter and ControlNet demonstrate that directly using LQ images or their features as guidance for diffusion sampling leads to incorrect codes and suboptimal restoration, since these methods were not developed for restoration tasks.
2.	The statement that diffusion models for BFR are trained on ImageNet appears to be inaccurate; these models are typically trained on LAION or its subsets.
3.	It is better to attribute the divergence of latent representations to the VAE, rather than to the inherent differences between LQ and HQ inputs.
4.	What qualitative improvements are observed when a Pruned Denoising UNet is added to Codebook Alignment?

**Questions:**

1.	Raises the intriguing hypothesis that the VAE fails to align LQ latent codes; however, the evidence provided does not yet constitute a convincing proof.
2.	The proposed method should preferably be compared with one-step BFR methods (e.g., InterLCM), instead of the fine-tuned version of OSEDiff on the FFHQ dataset.
3.	Claiming that using a small number of training samples constitutes a contribution is not well justified, as the training process is offline and performed only once (limiting the benefit of reduced training time), a dataset of 70K samples is already relatively small compared to the scale typically used in modern large models, using only 600 samples is likely to cause overfitting while 70K samples are less prone to it, and it remains unclear whether increasing the dataset size would improve reconstruction performance.

---

### Official Review · Reviewer_UrtC · 2025-10-31

**Soundness:** 3
**Presentation:** 3
**Contribution:** 3
**Rating:** 4
**Confidence:** 3

**Summary:**

The paper introduces LAFR, a lightweight framework for blind face restoration that addresses latent space misalignment between low-quality (LQ) and high-quality (HQ) facial images. By proposing a codebook-based alignment adapter and a multilevel restoration loss, the method achieves efficient adaptation of pretrained diffusion models with minimal data (0.9% of FFHQ) and parameters (7.5M). The approach demonstrates competitive performance on synthetic and real-world benchmarks while reducing training time by 70%.

**Strengths:**

1.     Efficiency and Innovation: The codebook-based alignment adapter is a novel solution to latent space misalignment, avoiding costly VAE retraining. The design is lightweight and modular, enabling effective domain adaptation with minimal parameters.

2.     Data Efficiency: The claim that facial images' structural regularity allows for effective training with only 600 images is well-supported by t-SNE analysis and ablation studies (Fig. 6, Tab. 11-12). This addresses a critical practical challenge in diffusion-based restoration.

3.     Comprehensive Evaluation: Extensive experiments on synthetic (CelebA-Test) and real-world (LFW, Wider, WebPhoto) benchmarks validate the method's robustness. The inclusion of efficiency metrics (Tab. 3) highlights practical advantages.

**Weaknesses:**

1.Limited quantitative performance: As shown in Tables 1 and 2, the proposed LAFR method underperforms compared to other approaches on nearly half of the evaluated metrics (e.g., DISTS↓, M-IQ↑, NIQE)

2.Presence of visual artifacts: The last row of Figure 4 indicates that LAFR tends to introduce noticeable visual artifacts in facial textures. This issue, though not universal, is observable in other test cases as well

3.Reproducibility and Implementation Clarity: The pruning strategy for the UNet (removing text/timestep embeddings) is described briefly. How this affects the diffusion process’s dynamics warrants more discussion.

4. Limitations of Structural Regularity: The method relies heavily on facial images' compact distribution (Fig. 10). While effective for frontal/near-frontal faces, performance on extreme poses or non-facial images is not evaluated. The appendix (Sec. A.4) acknowledges this but lacks empirical validation.

5.Baseline Comparisons: Comparisons with FaithDiff and OSDFace (Tab. 4) focus on alignment modules but do not ablate the full pipeline (e.g., LAFR vs. FaithDiff with identical training data).

**Questions:**

Please refer to the paper weakness

---

### Official Review · Reviewer_UVh1 · 2025-10-31

**Soundness:** 3
**Presentation:** 3
**Contribution:** 2
**Rating:** 2
**Confidence:** 3

**Summary:**

This paper proposes LAFR (Latent Alignment for Face Restoration), an efficient diffusion-based framework for blind face restoration that addresses the latent space misalignment between low-quality (LQ) and high-quality (HQ) images. LAFR introduces a lightweight codebook-based alignment adapter to align LQ latent representations with HQ distributions without retraining the original VAE, complemented by a multilevel restoration loss integrating identity embeddings and facial structural priors to preserve identity. By leveraging the structural regularity of facial images, the framework achieves state-of-the-art performance with only 0.9% of the FFHQ dataset (600 images) for fine-tuning, reducing training time by 70% and maintaining 7.5M trainable parameters. Extensive experiments on synthetic (CelebA-Test) and real-world (LFW-Test, Wider-Test) benchmarks validate its effectiveness in high-fidelity, identity-preserving restoration.

**Strengths:**

1. The codebook-based latent alignment adapter efficiently resolves LQ-HQ latent misalignment without modifying the pre-trained VAE, minimizing computational overhead compared to retraining-based alternatives.
2. The multilevel restoration loss (integrating appearance, identity, and structural supervision) ensures robust identity preservation, a critical requirement for face-specific restoration tasks.
3. Exceptional data and parameter efficiency, achieving competitive results with only 600 training images and 7.5M trainable parameters, reducing training time and resource demands.

**Weaknesses:**

1. The use of a very small training set (600 images) is advantageous in terms of computational efficiency, but this might limit the model's generalizability in scenarios with more varied or complex datasets, particularly when faces exhibit significant variations in expressions, lighting, or occlusions.
2. The novelty and contribution of the work are limited. The core innovation is restricted to the codebook-based latent alignment adapter for bridging LQ and HQ feature distributions, while the proposed multilevel restoration loss—integrating identity and facial structure constraints—relies on commonly used components (e.g., CLIP-based identity embeddings, 3D structural feature matching) that have been widely adopted in prior face restoration works. These loss designs lack sufficient originality to be considered distinct contributions, reducing the overall technical novelty of the framework.
3. The alignment adapter exhibits poor scalability, as evidenced by experimental results in Fig. 6. The figure shows that increasing the training image count (beyond 600 samples from FFHQ) does not yield proportional improvements in performance—with fluctuations and marginal gains observed even when training data is scaled up. This suggests inherent limitations in the adapter’s design: its reliance on a fixed HQ-learned codebook and shallow feature mapping fails to leverage additional training data effectively, indicating that the adapter cannot scale to capture more diverse LQ-HQ feature correlations, which may restrict its applicability to scenarios with larger or more varied datasets.

**Questions:**

See above

---

### Official Review · Reviewer_Mo7J · 2025-11-04

**Soundness:** 3
**Presentation:** 3
**Contribution:** 3
**Rating:** 6
**Confidence:** 4

**Summary:**

The paper proposes **LAFR**, a lightweight Blind Face Restoration (BFR) framework that keeps the original VAE frozen and adds a Latent Codebook Alignment Adapter to map *mismatched* latent codes of low-quality (LQ) faces toward the high-quality (HQ) latent distribution. A small LoRA finetune is applied to the diffusion U-Net, together with multi-level losses (pixel/perceptual, identity, structural) to preserve identity and structure. The authors claim competitive quality with only 7.5M trainable parameters and 0.9% FFHQ (≈600 images), and report 30% training-time savings. Experiments on synthetic (CelebA) and real sets (WebPhoto/LFW/WIDER, etc.) compare against FaithDiff, OSDFace, DiffBIR, and single-step baselines, with ablations.

**Strengths:**

Addresses the common “LQ→VAE latent mismatch” bottleneck via *codebook lookup + mapping* instead of retraining the VAE or adding heavy alignment blocks—reducing train/inference cost while stabilizing conditioning. Freezes the VAE; uses low-rank LoRA (e.g., rank=4) on selected U-Net convs; prunes unused text/time embeddings—yielding fewer parameters and lower latency than several strong baselines (including single-step diffusers). Multi-level losses and ablations indicate improved identity retention without sacrificing perceptual quality. t-SNE visualizations show latent alignment (LQ/HQ/Aligned separation), and the paper contrasts alignment strategies vs. prior work in parameters/latency/metrics.

**Weaknesses:**

1. Robustness under extreme degradations (heavy compression, strong motion blur, color shifts) and out-of-domain faces is not fully characterized.
2. Real data coverage is still limited; more device-/pipeline-specific degradations (smartphone ISP chains, night/IR, social-media recompression) would strengthen claims.
3. Beyond perceptual/quality metrics, standardized face-ID verification (e.g., ArcFace cosine vs. ground truth on synthetic pairs) is missing.
4. The stability/complexity of the codebook search (size, batch behavior, mixed precision) needs clearer analysis; comparisons with soft assignment or Top-k soft-NN would be helpful.

**Questions:**

1. How does LAFR behave under stronger JPEG, severe motion blur, non-Gaussian noise, or color cast? Please provide sensitivity curves and an alignment-success statistic (distance to HQ latent distribution).
2. On synthetic data with GT identities, report verification accuracy/cosine distributions (ROC/AUC) before/after alignment and with/without LoRA.
3. Detail codebook size, memory/throughput scaling, and runtime vs. Top-k soft-NN or Gumbel-Softmax alternatives.
4. Provide representative failure cases (attribute drift, artifacts), their frequency, and which components (codebook, LoRA location, loss weights) most affect them.

---

### Note · Authors · 2025-11-14

I have read and agree with the venue's withdrawal policy on behalf of myself and my co-authors.